# Influence of Grouting Sequence on the Correction Effect of Horizontal Tunnel Displacement by Grouting in Granite Residual Soil

**Min Zhu** [1,2]**, Changqing Xia** [1,2,*]**, Dengwei Chen** [3]**, Wei Chen** [4]**, Kun Hao** [5] **and Xiangsheng Chen** [1,2]

1   College of Civil and Transportation Engineering, Shenzhen University, Shenzhen 518061, China
2   Key Laboratory of Coastal Urban Resilient Infrastructures (MOE), Shenzhen University, Shenzhen 518061, China
3   China Railway Southern Investment Group Co., Ltd., Shenzhen 518060, China
4   Zhejiang University of Technology Engineering Design Group Co., Ltd., Hangzhou 310014, China
5   Electrical Engineering Co., Ltd. of CTCE Group, Bengbu 233000, China
*   Correspondence: xiacq09@szu.edu.cn

**Abstract:** Shield tunnels are vulnerable to large displacements induced by surrounding environmental changes. Grouting is an effective way to correct excessive shield tunnel displacement, while the grouting scheme design, especially the grouting sequence, is the key factor affecting the correction efficiency. A finite element simulation method considering the construction of multiple grouting zones is verified by the engineering case of Shenzhen Metro Line 1, and the influence of the grouting sequence on the correction effect on the horizontal tunnel displacement is further studied. The results show that the proper grouting efficiency of sandy clay, completely decomposed granite, and strongly decomposed granite is 5.5%, 2.8%, and 1.1%, respectively. The adjacent newly-built grouting zones are significantly constrained by the reinforcement created during the preceding grouting process. It is more efficient to correct excessive tunnel displacement in the "from far to near" sequence and excessive tunnel convergence in the "from near to far" sequence with increasing reinforcement stiffness. The correction effect improves greatly as the elastic modulus of the reinforcement increases up to 100 MPa.

**Keywords:** grouting; shield tunnel; soil-tunnel interaction; horizontal displacement; grouting sequence; granite residual soil

## 1. Introduction

In the past decades, a large number of metro tunnels have been built in coastal cities, and the shield tunnel structures are extremely susceptible to environmental changes due to complex geological conditions [1]. The excessive displacement of shield tunnels, attributed to a variety of factors including traffic load [2,3], nearby construction [4–6], and sudden surface surcharge [7], has resulted in a variety of tunnel damages, including segment dislocation, concrete damage, lining cracks, leakage, etc., which raises maintenance costs and affects the safe operation of the metro system [8].

Grouting has been proven to be a successful method for modifying differential displacement and relieving lining stress in existing metro tunnels. At present, there are two basic approaches: (1) internal grouting for the uplift of an operational tunnel [9–11], and (2) external grouting for the adjustment of differential displacement and the convergence of an existing tunnel. Because it has little to no influence on daily operations and imposes no time limits on construction, grouting from the outside has been employed successfully in many engineering projects [12–15]. At the same time, several academics have conducted theoretical studies on the grouting-induced tunnel response using the theoretical method [16], the finite element method [17], or the material point method [18].

The finite element method is effective in simulating the grout–soil–tunnel interaction. Grout injection into one single grouting hole is a small-scale process compared to the geometry size of shield tunnels. For actual engineering cases with hundreds of grouting holes, modeling every single grouting hole is not computationally efficient. The whole grouting process is often modeled on a more global scale, which is the soil-grout homogenization method [19]. Two different approaches are mainly used in the grouting simulation, including the application of internal pressure [20–22] or volumetric strain [23–25]. For the internal pressure method, the compressibility of soil elements treated by grouting is first reduced to a specified value. Internal pressure is applied to the grouting-treated elements until the volume expansion of these elements reaches the specific value. Then the stiffness of the treated elements is increased in order to simulate the reinforcement. For the volumetric strain method, a predefined volumetric strain determined by the injection volume is applied directly to the soil elements. However, there are usually a lot of grouting holes and a lengthy construction period in the engineering case. The physical properties are altered in the reinforcement area by previously constructed grouting holes, which will affect the upcoming grouting procedure. The impact of the building sequence is not taken into account when simulating the grouting procedure in the current research.

Based on an engineering case study for Shenzhen Metro Line 1, a numerical simulation approach of the grouting process that takes the grouting sequence into account was developed and verified. The effect of various grouting sequences on soil and tunnel displacement was then studied. The research results can provide theoretical support for the grouting scheme design and the optimization of similar projects.

## 2. Simulation of the Grouting Scheme with Multiple Grouting Zones

As shown in Figure 1, the entire reinforcement area is generally divided into multiple grouting zones, and each zone is formed by a certain number of grouting holes which are injected simultaneously. Because it is not economical or necessary to simulate every single grouting hole, a soil-grout homogenization method is adopted to capture the main characteristics of the grouting process to reduce computational costs. The grouting zone, which is mixed by the grout body and the soil within the diffusion range of grout veins, is set as the basic unit.

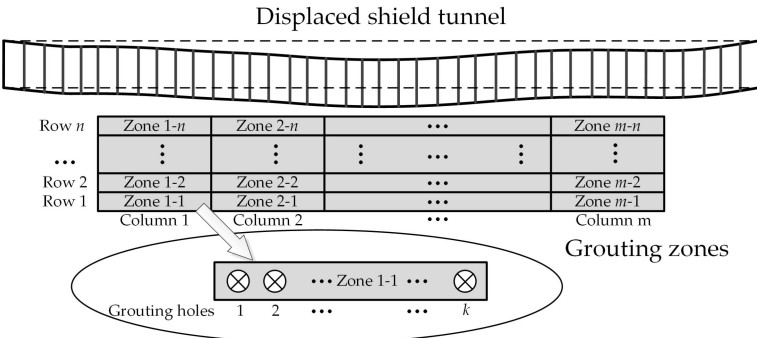

**Figure 1.** Plan view of a typical grouting scheme in the remediation project.

The following assumptions are proposed to simplify the simulation of one single grouting zone:

(1)　The length of the grout veins is limited due to the quick setting time of the cement and sodium silicate mixture, so the width of the grouting zone is set at half the designed row spacing.

(2)　A "prescribed pressure" method is used to simulate the grouting process of one grouting zone. The method is divided into two steps, including the expansion step and the reinforcement step. The expansion of the grouting zone is achieved by first reducing its compressibility to a very small value and then applying pressure to

the internal boundaries. In the reinforcement step, the mechanical properties of the grouting zone are enhanced and then the internal pressure is released.

In the expansion step, the internal pressure is gradually raised until the volume expansion ratio reaches a predetermined value, which is defined as:

$$\eta = \frac{V'_{\text{zone}} - V_{\text{zone}}}{V_{\text{zone}}} \tag{1}$$

where $\eta$ is the volume expansion ratio; $V_{\text{zone}}$ and $V'_{\text{zone}}$ are the total volumes of the grouting zone before and after grouting; $V_{\text{inj}}$ is the grout volume injected into the grouting zone. The grouting efficiency is defined as [16]:

$$\delta = \frac{V'_{\text{zone}} - V_{\text{zone}}}{V_{\text{inj}}} \tag{2}$$

where $\delta$ is the grouting efficiency. According to the field tests, the grouting-induced horizontal tunnel displacement decreases at first and tends to stabilize after 12 h [14]. In this paper, $\delta$ is defined as the final stable efficiency.

In the reinforcement step, the elastic modulus is calculated considering the proportion of the soil and the grout and the compressibility of the soil. The volume of soil and grout in the grouting zone is calculated, respectively, as:

$$V_{\text{soil}} = \frac{1 + e_1}{1 + e_0} V_{\text{zone}} \tag{3}$$

$$V_{\text{grout}} = \left(1 + \eta - \frac{1 + e_1}{1 + e_0}\right) V_{\text{zone}} \tag{4}$$

where $V_{\text{soil}}$ and $V_{\text{grout}}$ are the volumes of the soil and the grout after grouting; $e_0$ and $e_1$ are the void ratios of the soil before and after grouting; $V_{\text{zone}}$ is the total volume of the grouting zone.

The elastic modulus of the reinforcement after grouting $E$ is calculated by:

$$E = \frac{V_{\text{soil}} E_{\text{soil}} + V_{\text{grout}} E_{\text{grout}}}{V_{\text{soil}} + V_{\text{grout}}} = \frac{\frac{1 + e_1}{1 + e_0} E_{\text{soil}} + \left(1 + \eta - \frac{1 + e_1}{1 + e_0}\right) E_{\text{grout}}}{1 + \eta} \tag{5}$$

where $E_{\text{soil}}$ is the elastic modulus of soil after grouting; $E_{\text{grout}}$ is the elastic modulus of the grout mixture.

Due to the reinforcement of the grouting zones, the impact of the earlier-constructed grouting zones on the later-constructed grouting zones cannot be ignored in the remediation project. As a result, grouting is simulated zone by zone based on the planned construction sequence. Every single grouting zone employs the "prescribed pressure" method, and Equation (5) is used to determine the elastic modulus of the grout-soil reinforcement for the previously constructed grouting zones.

## 3. Engineering Verification

### 3.1. Engineering Background

The section of Shenzhen Metro Line 1 from Liyumen station to Qianhaiwan station is located in the coastal reclamation area of Qianhai Bay in Nanshan District, Shenzhen, China. The up-track tunnel was damaged due to the sudden drop in groundwater level caused by a nearby foundation excavation. The maximum accumulated horizontal displacement and settlement of the up-track tunnel finally reached 34.8 mm and 76.8 mm, respectively, which caused severe damage to the shield tunnel, including segment dislocation, leakage, spalling, and cracks on the track bed. The details can be seen in reference [13].

Figure 2 shows the design of the remediation scheme. To release the lining stress and reduce the excessive tunnel displacement, the ground soil above the up-track tunnel was

removed by 2 m at first, and the grouting technique was adopted to correct excessive tunnel deformation. Eight rows of grouting holes were arranged in a quincuncial shape, including six rows to the south of the damaged up-track tunnel and two rows between the up-track and the down-track tunnel. Each row of grouting holes was divided into seven grouting zones, and the injection sequence was every other hole from the central zone (Zone C-1) to both sides. The sleeve valve pipe grouting method was used, and the total height of the grouting zone was 9 m. The grout mixture was composed of an equal volume of cement with a water-to-cement ratio of 1:1 and sodium silicate of 39°Bé. The gelation time of the grout mixture was 111 s. Grouting pressure was set at 0.3–0.5 MPa, and the designed total injection volume per hole was 3 m³.

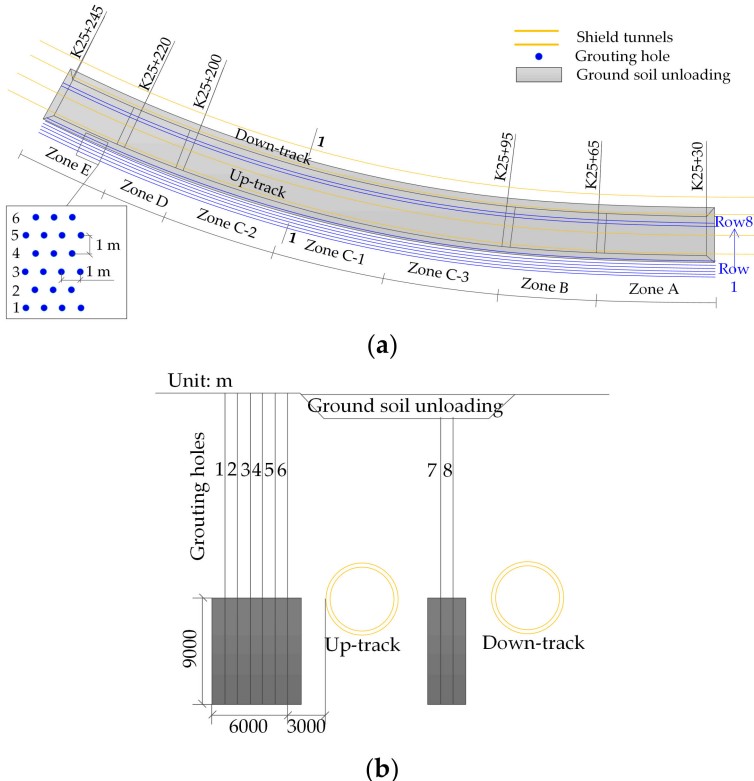

(**a**)

(**b**)

**Figure 2.** Scheme of the remediation project. (**a**) Plan view; (**b**) typical profile.

Figure 3 shows the geological profile. The physical and mechanical properties of the soil are listed in Table 1. The soil layers from top to bottom are the fill layer, the muddy clay layer, the silty clay layer, the sandy clay layer, and the Yanshanian granite layers with different weathering degrees. The soil layers of the grouting zones are mainly the sandy clay layer (Zone A, B, C-1, and C-3) and the decomposed granite layer (Zone D and E). The soil layers change from sandy clay to decomposed granite in Zone C-2. The groundwater level is 3 m below the ground surface.

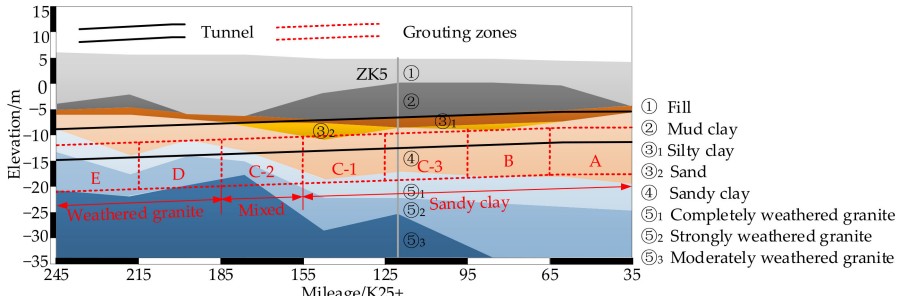

**Figure 3.** Geological profile.

**Table 1.** Physical and mechanical properties of the soil.

| Soil | $w$/% | $\rho$/g/cm$^3$ | $G_s$ | $e$ | $w_L$/% | $w_P$/% | $I_P$ | $N'$ |
|---|---|---|---|---|---|---|---|---|
| ② | 24.20 | 1.91 | 2.67 | 0.74 | / | / | / | 14.0 |
| ② | 86.90 | 1.49 | 2.66 | 2.44 | 60.50 | 35.00 | 25.50 | 1.1 |
| ③₁ | 20.70 | 2.01 | 2.68 | 0.61 | 33.70 | 19.80 | 13.90 | 10.8 |
| ③₂ | 24.30 | 1.97 | 2.69 | 0.70 | 38.00 | 22.40 | 15.60 | 17.2 |
| ④ | 31.30 | 1.82 | 2.67 | 0.92 | 43.30 | 25.70 | 17.60 | 15.7 |
| ⑤₁ | 22.30 | 1.89 | 2.67 | 0.74 | 36.40 | 21.70 | 14.70 | 39.4 |
| ⑤₂ | 21.40 | 1.89 | 2.67 | 0.72 | 35.50 | 21.20 | 14.30 | 58.7 |

Note: $w$ = water content; $\rho$ = natural density; $G_s$ = specific gravity; $e$ = void ratio; $w_L$ = liquid limit; $w_P$ = plastic limit; $I_P$ = plastic index; $N'$ = modified SPT value.

### 3.2. Establishment of 3D Finite Element Model

1. Finite element model

A 3D finite element model is established based on the aforementioned site conditions, as shown in Figure 4. The geometric size of the model is 280 m × 250 m × 36 m. The radius of the up-track and down-track tunnels is 415 m and 400 m, respectively. The distribution of soil layers is simplified while the main characteristics are retained. Above the up-track tunnel is a 15 m wide by 2 m deep unloading zone. 174,150 elements and 271,591 nodes make up the entire model. The horizontal displacement is restricted along four lateral boundaries, and the bottom is fixed. A free drainage boundary is established at the upper boundary. The outer diameter of the shield tunnel is 6 m, and the thickness of the linings is 0.30 m.

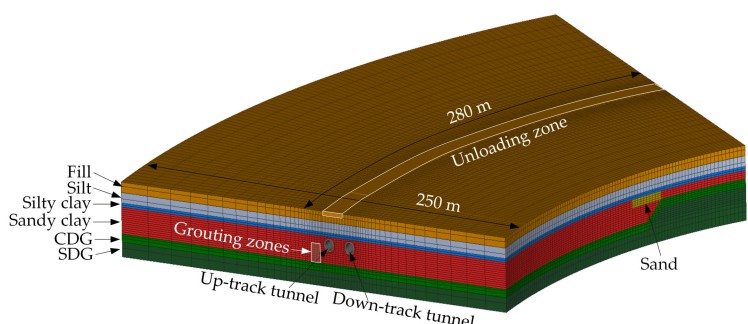

**Figure 4.** Finite element model.

2. Parameters

Typical parameters of different soil layers were obtained from field tests and laboratory triaxial tests [25]. The modified Cambridge model is used to describe the mechanical behavior of the soil layers of muddy clay and silty clay. The Mohr–Coulomb model is used in the soil layers of sandy clay, completely decomposed granite (CDG), strongly decomposed granite (SDG), and sand. The mechanical parameters of different soil layers are listed in Tables 2 and 3.

**Table 2.** Parameters of the modified Cam–Clay model.

| Soil | $k$/m/s | $\kappa$ | $\nu$ | $\lambda$ | $M$ | $e_1$ |
|---|---|---|---|---|---|---|
| Muddy clay | $1.2 \times 10^{-9}$ | 0.0157 | 0.30 | 0.1838 | 1.12 | 2.24 |
| Silty clay | $5.8 \times 10^{-8}$ | 0.0106 | 0.26 | 0.0626 | 1.05 | 0.93 |

Note: $k$ = permeability coefficient; $\kappa$ = log bulk modulus; $\nu$ = Poisson's ratio; $\lambda$ = void ratio; $M$ = stress ratio; $e_1$ = initial void ratio (used to determine the initial yield surface).

**Table 3.** Parameters of the Mohr–Coulomb model.

| Soil | $k$/m/s | $E_s'$/MPa | $\nu$ | $c'$/kPa | $\varphi'$/° |
|---|---|---|---|---|---|
| Sandy clay | $9.7 \times 10^{-8}$ | 15.7 | 0.25 | 20 | 28 |
| Sand | $5.2 \times 10^{-5}$ | 30.0 | 0.30 | 0 | 30 |
| CDG | $1.0 \times 10^{-5}$ | 43.3 | 0.23 | 35 | 30 |
| SDG | $1.9 \times 10^{-5}$ | 82.2 | 0.20 | 45 | 35 |

Note: $k$ = permeability coefficient; $E_s'$ = effective elastic modulus; $\nu$ = Poisson's ratio; $c'$ = effective cohesion; $\varphi'$ = effective friction angle; CDG is the completely decomposed granite and SDG is the strongly decomposed granite.

The elastic modulus of the shield tunnel lining is 34.5 GPa and the Poisson's ratio is 0.2. The shield tunnel is regarded as homogenous and the equivalent longitudinal bending stiffness and the equivalent transverse bending stiffness are respectively set as 0.17 and 0.7. The normal contact between the soil and the tunnel is hard contact, and the tangential friction coefficient is 0.49 [26,27].

3.    Grouting scheme

Considering that there are 56 grouting zones in total, it is difficult to simulate every single grouting zone. As shown in Figure 5, the whole grouting process is simplified into four main stages. The grouting zones in the same stage were constructed within one week, so they are simulated simultaneously in one numerical analysis step.

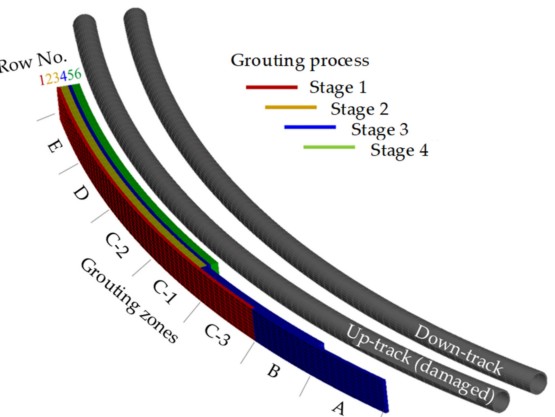

**Figure 5.** Stages of the grouting process.

The grouting process of each stage is simulated by the method proposed in the previous section. The back-analysis of Stage 2 was carried out to determine the proper volume expansion ratio $\eta$ of different soil layers. The proper volume expansion ratio for the sandy clay, CDG, and SDG is 2.0%, 1.0%, and 0.4%, respectively, as shown in Table 4. Considering that the average measured injection volume for each grouting hole is 3.3 m³ and the distance between adjacent grouting holes is 1 m, the grouting efficiency of the sandy clay, CDG, and SDG is 5.5%, 2.8%, and 1.1%, respectively. The grouting efficiency is relatively low compared to similar projects and decreases significantly as the weathering degree decreases due to the larger permeability coefficients and more developed fissures of CDG or SDG. The flow of seawater also aggravates the loss of grout.

The average void ratio of the soil layers before and after grouting was obtained by borehole sampling in the field. The soil is compressed after grouting, and the elastic modulus of the reinforcement is calculated by Equation (5). The shear strength parameters are increased by 60–150% on average after grouting, according to laboratory tests [28,29].

**Table 4.** Parameters of the reinforcement of different soil layers.

| Soil | $\eta$/% | $\delta$/% | $E_s'$/MPa | $E_g$/MPa | $e_0$ | $e_1$ | $E$/MPa |
|---|---|---|---|---|---|---|---|
| Sandy clay | 2.0 | 5.5% | 15.7 | | 0.92 | 0.89 | 67.5 |
| CDG | 1.0 | 2.8% | 43.3 | 1500 | 0.74 | 0.68 | 107.5 |
| SDG | 0.4 | 1.1% | 82.2 | | 0.72 | 0.70 | 104.3 |

Note: $r$ = volume expansion ratio; $E_s'$ = effective elastic modulus of the soil; $E_g$ = elastic modulus of the grout mixture; $e_0$ = void ratio of the soil before grouting; $e_1$ = void ratio of the soil after grouting; $E$ = elastic modulus of the reinforcement.

### 3.3. Tunnel Displacement

Figure 6 shows the grouting-induced horizontal displacement of measuring point 1, located on the left waist of the up-track tunnel. Because Stage 2 is the back-analysis step, the tunnel displacement of the field measurements and the numerical analysis are in good agreement based on the proper determination of the volume expansion ratio of different soil layers, as shown in Figure 6a. Due to various grouting efficiencies in different soil layers, the measured tunnel displacement reaches its peak value in grouting Zone C-1 and decreases sharply in grouting Zones C-2, D and E. Compared with the results of the numerical simulation, the measured displacement curve has a stepped shape, possibly due to the grouting sequence of different zones in the same stage and the uneven spatial distribution of soil layers. Stage 3 and Stage 4 are the verification steps, and the measured tunnel displacement is still consistent with the tunnel displacement of numerical analysis, as shown in Figure 6b,c. According to the field measurements, the construction of grouting holes in Row 5 and Row 6 causes sudden changes in the tunnel displacement curve at K25 + 94.5, K25 + 193.5, and K25 + 238.5. The possible reasons are as follows: (1) The shield tunnel had been severely damaged due to nearby foundation excavation, and the longitudinal joints were possibly weakened. In the numerical simulation, the shield tunnel is simplified as a homogenous tube, and the nonuniformity of the tunnel structure is not considered. (2) The locations where sudden changes occur are close to the boundaries of the grouting zones. In the same row, the grouting sequence is from the center to both sides. In the same stage, the grouting process is not considered in the numerical simulation.

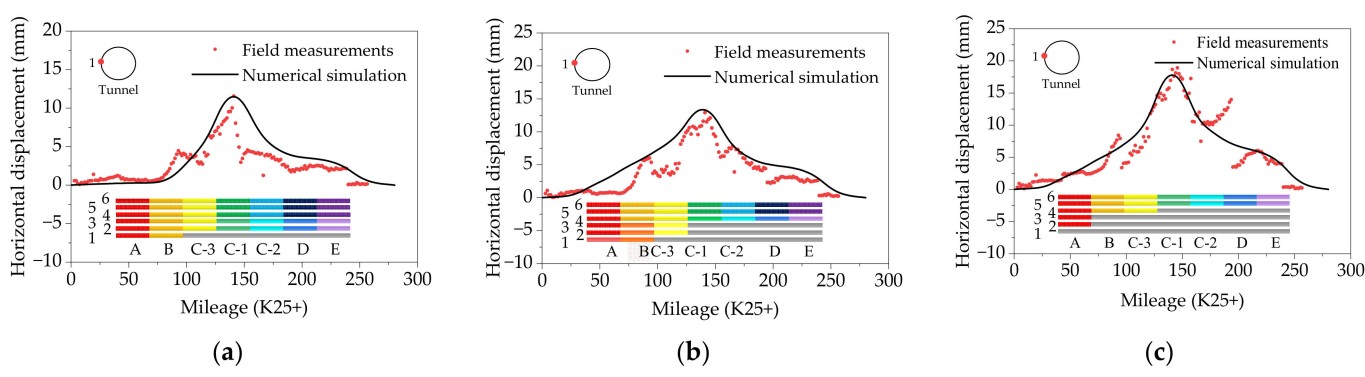

**Figure 6.** Grouting-induced horizontal tunnel displacement. (**a**) Stage 2; (**b**) Stage 3; (**c**) Stage 4.

## 4. Influence of the Grouting Sequence

### 4.1. Establishment of 3D Finite Element Model

A simplified 3D numerical model is established to investigate the influence of existing reinforcement on the following grouting process, as shown in Figure 7. The geometric size of the new model is 210 m × 60 m × 30 m. The outer diameter of the shield tunnel is 6 m, and the thickness of the lining is 0.3 m. The buried depth of the shield tunnel is 12 m. The geometric size of the grouting zone is 30 m × 1 m × 9 m. The grouting zone under construction is 4 m on the left side of the left tunnel boundary. The horizontal displacement of the four boundaries is constrained, and the bottom is fixed. Free drainage is allowed at

the upper boundary. The mechanical parameters of the soil are set the same as the sandy clay in Table 4. The structural parameters of the shield tunnel are the same as in Section 3.2.

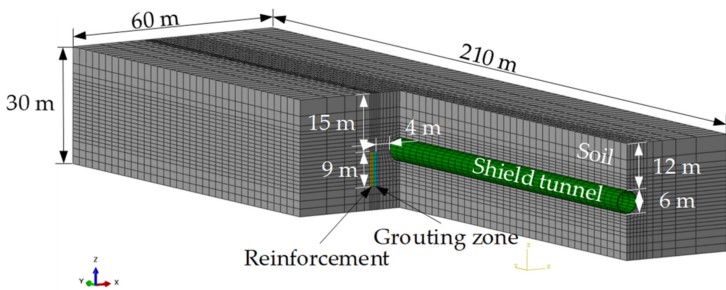

**Figure 7.** Finite element model for the parametric analysis.

As shown in Figure 8, three typical cases of different grouting sequences of multiple grouting zones are considered in the numerical analysis, respectively named "from far to near" (Case 1), "from one side to the other" (Case 2), and "from near to far" (Case 3). The volume expansion ratio of the grouting zone in all cases is set at 2%. The elastic modulus of the reinforcement ranges from 10 MPa to 1000 MPa.

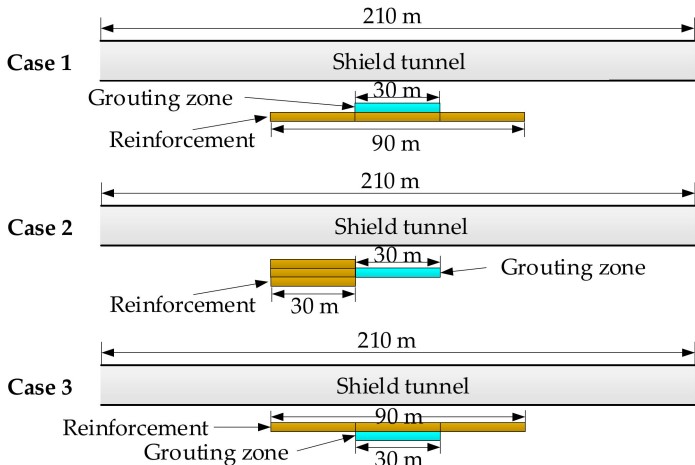

**Figure 8.** Cases of different grouting sequences.

### 4.2. Influence of Grouting Sequence on the Tunnel Deformation

Figure 9 shows the horizontal displacement of the tunnel axis under different grouting sequences. In all three cases, the longitudinal influence range of the grouting process on the shield tunnel is about three times the length of the grouting zone. The peak displacement of the tunnel axis increases with increasing the elastic modulus of the reinforcement in the "from far to near" sequence (Figure 9a). The larger stiffness of the reinforcement results in a better remediation effect of displaced shield tunnels in the "from far to near" construction mode, because the stiffer reinforcement provides stronger boundary constraints. The tunnel displacement on the reinforcement side gradually decreases as the elastic modulus of the reinforcement increases in the "from one side to the other" sequence (Figure 9b). The peak value and the tunnel displacement away from the reinforcement side are not significantly affected. The tunnel displacement increases initially and then decreases in the "from near to far" sequence (Figure 9c), reaching a maximum when the elastic modulus of the reinforcement is 100 MPa.

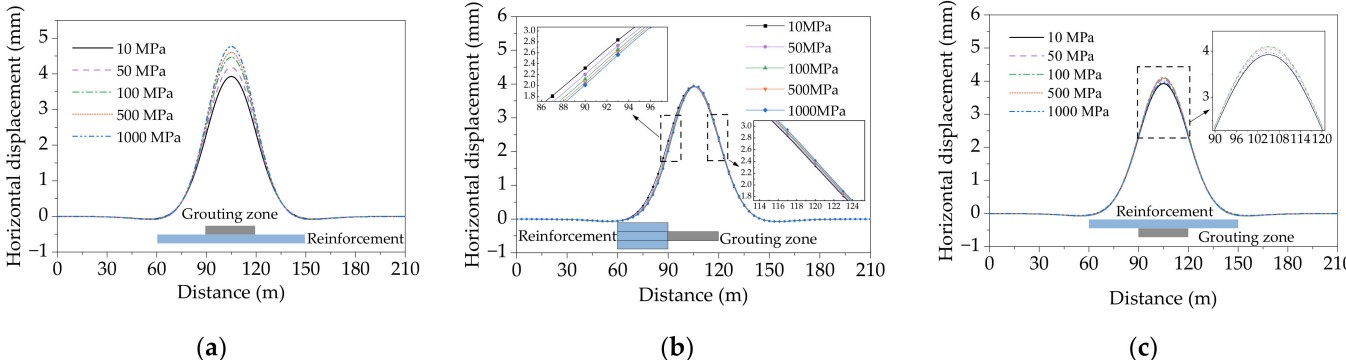

**Figure 9.** Grouting-induced horizontal tunnel displacement. (**a**) From far to near (Case 1); (**b**) from one side to the other (Case 2); (**c**) from near to far (Case 3).

The tunnel deformation is composed of translation and shape change [13], which is represented by the tunnel displacement and the horizontal convergence, respectively. Figure 10 shows the deformation of the central tunnel section where the maximum tunnel displacement occurs. For the "from far to near" sequence (Figure 10a), the horizontal tunnel displacement and the horizontal convergence rapidly increase at first, with the increase of the elastic modulus of the reinforcement. When the elastic modulus exceeds 100 MPa, the growth rate of tunnel displacement slows down sharply, and the horizontal convergence change tends to be stable. With the increase in the elastic modulus of the reinforcement, the ratio of $\Delta D/s$ reduced from 0.528 to 0.465, indicating that more translation of the tunnel section takes place instead of shape change. It is more effective to reduce excessive tunnel displacement rather than horizontal convergence by increasing the stiffness of the reinforcement in the "from far to near" sequence.

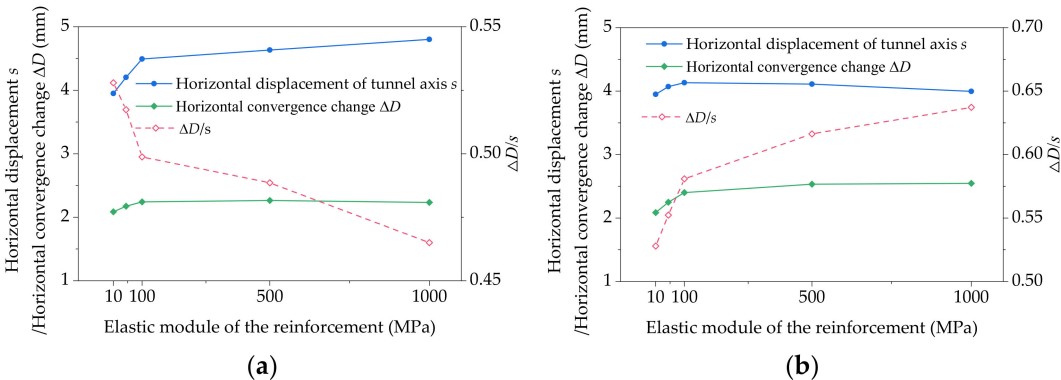

**Figure 10.** Tunnel displacement and convergence under different grouting sequences. (**a**) From far to near (Case 1); (**b**) from near to far (Case 3).

For the "from near to far" sequence (Figure 10b), the horizontal displacement of the tunnel axis reaches a maximum when the elastic modulus of the reinforcement increases to 100 MPa. The growth rate of the horizontal tunnel convergence is fast from 10 MPa–100 MPa and slows down from 100 MPa–1000 MPa. The ratio of $\Delta D/s$ increases from 0.528 to 0.637, indicating that more shape change in the tunnel section occurs. Through the grouting sequence of "from near to far", increasing the stiffness of the reinforcement is more effective in reducing large tunnel convergence.

### 4.3. Influence of Grouting Sequence on the Soil Deformation

Figure 11 shows the soil displacement parallel to the grouting zone under different grouting sequences. For the "from far to near" sequence (Figure 11a) and the "from near to far" sequence (Figure 11c), the soil displacement on the reinforcement side decreases and

the soil displacement on the opposite side increases with the increase of the reinforcement stiffness. For the "from one side to the other" sequence (Figure 11b), the soil displacement on the reinforcement side is strongly constrained by the reinforcement. Figure 12 shows the soil displacement perpendicular to the grouting zone. With the increase in the reinforcement stiffness, the soil displacement on the reinforcement side decreases and the soil displacement on the opposite side increases. Soil displacement decreases when the distance from the grouting zone increases.

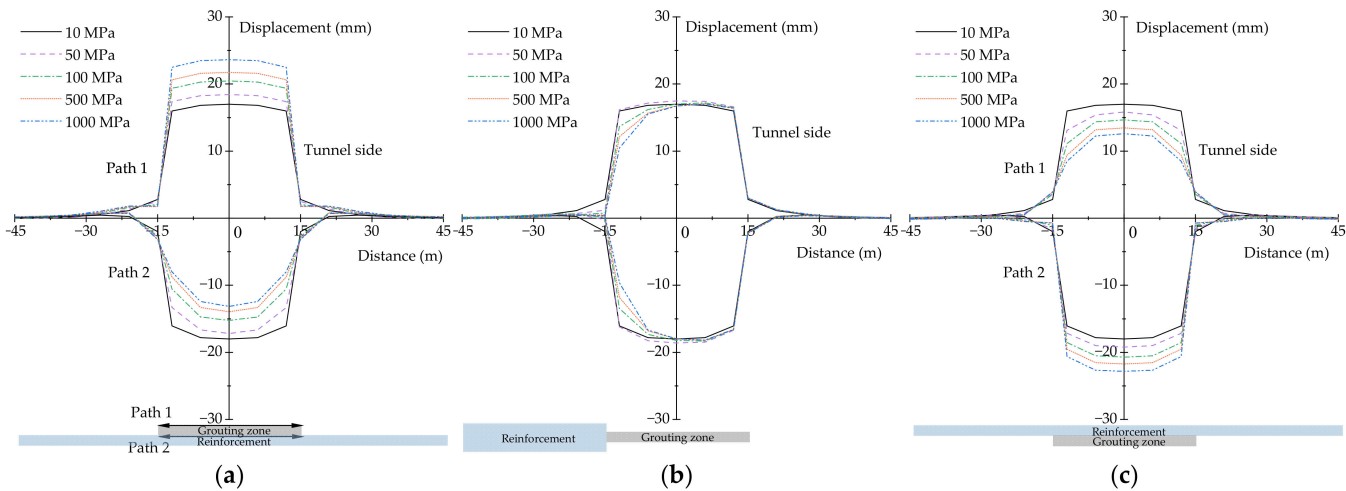

**Figure 11.** Soil displacement of the grouting zone. (**a**) Case 1; (**b**) Case 2; (**c**) Case 3.

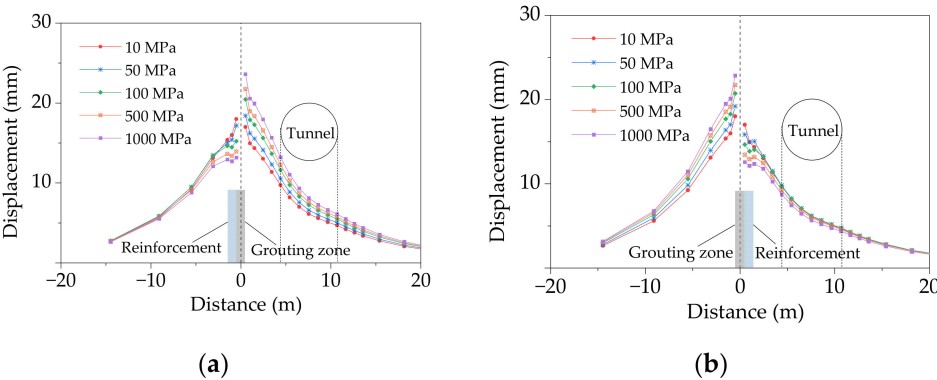

**Figure 12.** Soil displacement perpendicular to the grouting zone. (**a**) From far to near (Case 1); (**b**) from near to far (Case 3).

## 5. Conclusions

In this paper, a numerical simulation method for the grouting process that considers the grouting sequence was proposed and verified, and the influence of different grouting sequences on the tunnel and soil displacement was studied. The conclusions are as follows:

1.  Taking the grouting zone as the basic unit, the grouting process is simulated by the soil-grout homogenization method. The proper grouting efficiency of sandy clay, CDG, and SDG is 5.5%, 2.8%, and 1.1%, respectively. The grouting efficiency decreases significantly as the weathering degree decreases.
2.  The reinforcement formed in the previous grouting process has strong constraints on the adjacent newly-built grouting zones. It is more efficient to correct excessive tunnel displacement in the "from far to near" sequence and excessive tunnel convergence in the "from near to far" sequence with increasing reinforcement stiffness. The correction effect improves significantly as the elastic modulus of the reinforcement increases from 10 MPa to 100 MPa.

3. According to the comparison between the field measurement and the numerical analysis, a more precise tunnel model with segments and joints, the initial tunnel condition in accordance with the actual situation, and a more detailed simulation of the grouting process are key to improving the accuracy of the numerical analysis.

The research on the tunnel displacement correction effect under different grouting sequences can provide theoretical support for the optimal design of grouting schemes in similar projects. However, it is found in the field construction that in the weathered granite layer with more developed fissures, grout loss is a serious concern due to the influence of groundwater flow, thus better water stop measures are required before grouting construction.

**Author Contributions:** Conceptualization, M.Z., C.X. and X.C.; methodology, M.Z. and C.X.; software, M.Z. and C.X.; validation, D.C. and M.Z.; formal analysis, M.Z.; investigation, W.C., C.X. and M.Z.; resources, D.C. and C.X.; data curation, D.C.; writing—original draft preparation, M.Z.; writing—review and editing, M.Z., W.C., K.H. and X.C.; visualization, D.C., K.H. and C.X.; supervision, M.Z.; project administration, M.Z., C.X. and W.C.; funding acquisition, M.Z., C.X. and X.C. All authors have read and agreed to the published version of the manuscript.

**Funding:** This research was funded by the National Natural Science Foundation of China, grant number 52008263, grant number 52108329, grant number 52090084; the China Postdoctoral Science Foundation, grant number 2021T140475; the Shenzhen Science and Technology Program, grant number KQTD20200909113951005; and the 2020 Science and Technology Development Project of China Railway Group Limited, grant number 2020-key project-14.

**Institutional Review Board Statement:** Not applicable.

**Informed Consent Statement:** Not applicable.

**Data Availability Statement:** Data is contained in the article.

**Conflicts of Interest:** The authors declare no conflict of interest.

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
