# Peer review of "Influence of Grouting Sequence on the Correction Effect of Horizontal Tunnel Displacement by Grouting in Granite Residual Soil"

_buildings, doi:10.3390/buildings12101655_

Round 1

Reviewer 1 Report

General Considerations:

·               The topic matter of the proposal addresses a very common problem in the study of metro tunnel design, and just because of the relevance of the proposed case study, it deserves to be considered for a high impact publication.

·               The manuscript, which counts all its sections, has a total length of 12 pages. For this reason, and because of the nature of the research it provides in relation to the contents presented, it can be considered a good work.

·               The document includes a total of 29 references, of which 36% are publications produced in the last 5 years, 38% in the last 5-10 years, and 26% are more than 10 years old, implying a total percentage of 74 % recent references. In this way, the total number of references used can be considered appropriate.

·               The reference number 29 has not been possible to check online; this should be improved or changed.

·               The spelling of some word should be checked, the division of words between lines is, in some cases, incorrect, for instance, lines 50-51 , 53-54, 64-65, 128-129, 195-196, 205-206, 237,238, 262-263, and many others.

Title, Abstract, and Keywords:

·               After analysing the contents of the article in detail, the title " Influence of Grouting Sequence on the Correction Effect of Horizontal Tunnel Displacement by Grouting in Granite Residual Soil" and the abstract are both correct.

·               The keyword section is correct.

Section 1: Introduction

·               The artificial value mentioned in line 57 should be better described.

·               In line 64 there is a wrong word “ha1ve” not in the dictionary.

2. Simulation of the grouting scheme with multiple grouting zones

·       Figure 1 can be better described; is it a vertical or horizontal cross section?

3. Engineering verification

·       The description of the soil layers could be improved and the position of the water table could be included to at least check whether the pore water pressure´ has any influence on the calculations.

4. Influence of the grouting sequence

·       Line 237 The word cannot be divided into two like this “exist-ing”

·               The text in Figure 15 has a margin for improvement; they could be the same size as in Figure 14, they are difficult to read.

5. Conclusions

·               The conclusions section could provide reflections on the replicability of the method in a similar situation or soil the possible effect on the calculations of the existence and position of the groundwater table and therefore of the water pressure.

·               More emphasis could also be placed on the degree of novelty of the proposal and the general advantages it provides.

Final evaluation

In summary, the research is very interesting, but the current document has several weaknesses that must be strengthened to obtain a documentary result that is equal to the value of the publication.

Reviewer 2 Report

This research presents a case study on finite element simulation on Shenzhen Metro Line 1; and the influence of grouting sequence on the correction effect of horizontal tunnel displacement was further investigated. This research would be interesting for the readers of the journal and can be accepted after a through grammar and spell check. The authors are suggested to do a comprehensive proof-read. Some sentences are vague with errors.

Author Response

Thanks to the reviewer for the comments. The authors have checked the grammar and spelling in the manuscript. Please see the revised manuscript for the detailed changes.